# The Impact of Dietary Intake of Furocoumarins and Furocoumarin-Rich Foods on the Risk of Cutaneous Melanoma: A Systematic Review

**DOI:** 10.3390/nu17081296

**Published:** 2025-04-08

**Authors:** Isabelle Kaiser, Anja Rappl, Lena S. Bolay, Annette B. Pfahlberg, Markus V. Heppt, Olaf Gefeller

**Affiliations:** 1Department of Medical Informatics, Biometry and Epidemiology, Friedrich-Alexander-Universität Erlangen-Nürnberg, 91054 Erlangen, Germany; isabelle.kaiser@fau.de (I.K.); anja.rappl@fau.de (A.R.); lena.heiss@fau.de (L.S.B.); annette.pfahlberg@fau.de (A.B.P.); 2Department of Dermatology, Universitätsklinikum Erlangen, Friedrich-Alexander-Universität Erlangen-Nürnberg, 91054 Erlangen, Germany; markus.heppt@uk-erlangen.de; 3Comprehensive Cancer Center Erlangen-European Metropolitan Area of Nuremberg (CCC ER-EMN), 91054 Erlangen, Germany

**Keywords:** citrus fruits, cutaneous melanoma, diet, furocoumarin, grapefruits, ultraviolet radiation

## Abstract

Background/Objectives: Furocoumarins, chemical compounds found in many plant species, have a photosensitizing effect on the skin when applied topically and, by interacting with ultraviolet radiation (UVR), stimulate melanoma cells to proliferate. Whether dietary intake of furocoumarins acts as a melanoma risk factor has been investigated in several epidemiological studies, which are synthesized in our systematic review. Methods: The study protocol was registered with PROSPERO (registration number: CRD42023428596). We conducted an in-depth literature search in three databases coupled with forward and backward citation tracking and expert consultations to identify all epidemiological studies, irrespective of their design, addressing the association between a furocoumarin-containing diet and melanoma risk. We extracted information on the study details and results in a standardized manner and evaluated the risk of bias of the results using the Joanna Briggs Institute’s critical appraisal tools. Results: We identified 20 publications based on 19 different studies providing information on the association between dietary furocoumarin intake and melanoma risk. We refrained from a meta-analytical synthesis of the results because of the large heterogeneity in exposure assessment, operationalization of furocoumarin intake in the analyses, and analytical methods of the studies. In a qualitative synthesis, we found moderate evidence supporting the notion that dietary furocoumarin intake at higher levels acts as a risk factor for cutaneous melanoma. Conclusions: Our systematic review provides an overview of the current epidemiological evidence, but it could not clearly answer whether and to what extent dietary furocoumarin intake increases melanoma risk. Future epidemiological analyses focusing on this topic require more comprehensive dietary and UVR exposure data to better characterize the individual total furocoumarin intake and its interplay with UVR exposure patterns.

## 1. Introduction

Uncontrolled proliferation of melanocytes located in the basal layer of the skin’s epidermis leads to cutaneous melanoma (CM), a malignant tumor representing a form of skin cancer that causes more than 55,000 deaths globally each year when diagnosed at an advanced stage [1,2]. Approximately 325,000 new cases of CM occurred worldwide in 2020, with large geographic variations in incidence across countries and world regions [2]. Historically, CM has been considered a rare disease, but the incidence of CM in fair-skinned populations has rapidly increased over the past decades [3]. Epidemiologic studies have identified changes in exposure to ultraviolet (UV) radiation (UVR) as the main driver of this development [4]. The role of dietary factors in the development of CM has also been investigated in numerous studies [5,6]. For example, caffeine consumption was found to be inversely associated with CM risk in a dose-response meta-analysis of prospective cohort studies [7]. However, the results of epidemiologic studies on most dietary factors are inconsistent. Overall, the current consensus is that dietary factors play only a subordinate role in CM risk and are not incorporated in risk prediction models for CM [8,9,10].

During the last two decades, particular attention has been paid to the potential role of dietary furocoumarins in the development of CM [11]. Furocoumarins are found in many plant species that synthesize furocoumarin via the fusion of coumarin with a furan ring [12]. Different furocoumarin isomers are generated depending on the position of the furan ring. The photosensitizing properties of furocoumarins on the skin when applied topically have been known for almost a century [13] and have been exploited for therapeutic purposes in the treatment of skin diseases, such as, e.g., psoriasis and vitiligo, by combining psoralen, one of the furocoumarin isomers, with UV-A radiation in the so-called PUVA therapy [14]. Follow-up of patients treated with PUVA has shown that the risk for CM and keratinocyte cancers is substantially increased in these patients [15]. The effect of dietary furocoumarin intake on the risk of CM has been investigated in a number of studies, most of which have focused on the consumption of citrus fruits because of their high furocoumarin content. A comprehensive overview of all epidemiologic studies providing data on the association between dietary furocoumarin consumption and CM risk is currently lacking.

The aim of our work is to fill this gap by synthesizing all available information on the impact of dietary intake of furocoumarins and furocoumarin-rich foods on the risk of CM. We systematically searched the scientific literature, compiled the first comprehensive review of all epidemiologic studies on this topic, regardless of their design, and evaluated the current evidence on the effect of dietary furocoumarin intake on CM risk.

## 2. Materials and Methods

We developed a study protocol that was registered with PROSPERO (registration number: CRD42023428596). The systematic review was conducted and reported according to the PRISMA guideline [16].

### 2.1. Eligibility Criteria

The eligibility criteria were defined according to the PECOS scheme, which is a modified version of the PICO scheme that is more suitable for epidemiologic studies, as it contains the aspect “exposure” instead of “intervention” [17]. We included all observational epidemiologic studies that investigated the association between furocoumarin-containing foods and CM. Eligible furocoumarin-containing foods are listed in Table 1. We excluded animal studies, but further restrictions regarding the study population were not made. Furthermore, only studies with full texts written in English or German were included.

### 2.2. Search Process

We conducted an electronic literature search across three different databases (PubMed/Medline, Scopus/EMBASE, and Web of Science—Core Collection). In addition, we employed the forward snowballing technique for nine studies relevant to this topic that were identified during the database search (see Appendix A). Forward snowballing is an efficient search approach that investigates citations to specific reference papers and, thus, looks forward in time when performing a search among citations. We used the Scopus database for this purpose. Additionally, the references in the publications identified through the electronic searches that met our inclusion criteria were manually screened for relevant publications. We also contacted dermato-oncological experts and asked for further references.

Before implementation, the search process was reviewed by members of the team who were not involved in its original development. This peer review was performed in accordance with the PRESS guideline [18].

The search strings used to search the three databases can be found in the Appendix A. We conducted the original search in April 2023 and updated it in January 2025.

### 2.3. Study Selection

The screening and selection process for all eligible studies comprised two phases. After removing duplicates, the titles and abstracts of all identified publications were evaluated for eligibility. Four reviewers were involved in the screening process, with the corpus of identified references divided up so that each reference was screened independently by two individuals. All publications deemed potentially eligible by at least one reviewer were included in the full-text screening. Thus, only publications that were judged as not relevant by both reviewers were excluded, which ensured a high sensitivity of the search procedure. In the second full-text screening phase, the eligibility of each publication was evaluated by two reviewers independently. Discrepancies in decisions were resolved by discussion and, in rare cases of sustained disagreement, by involving an independent third reviewer.

### 2.4. Data Extraction

Similar to the procedure employed for selecting the studies, the data extraction was also conducted in accordance with the four-eyes principle in a mutually blinded fashion. Discrepancies between the two data extractions were resolved by discussion, potentially involving a third reviewer. A data extraction form was developed and pretested. For each publication, information was extracted on the relevant study characteristics, such as publication year, data collection period, study design, sample size, and method of selecting participants. Additionally, details were gathered on the type and quantification of dietary exposure, the statistical analysis employed, and the results reported for the association between furocoumarin-containing foods and CM risk. Since the hypothesized link between furocoumarin intake and CM risk involves furocoumarin–UVR interaction [19], we further assessed how information about UVR exposure was gathered and taken into account when quantifying the risk of CM due to furocoumarin-containing diet.

### 2.5. Risk of Bias Assessment

The risk of bias (ROB) of all studies included in the systematic review was assessed independently by three reviewers using the Joanna Briggs Institute (JBI) checklists. As studies with different study designs had to be evaluated, the corresponding JBI checklist was employed for each study design. The checklists included those for case-control, cohort, and analytical cross-sectional studies, with each comprising between 8 and 11 items related to participant selection, exposure and outcome assessment, and statistical analysis, which are either answered with yes or no depending on whether the item was met or not. If the information reported in the study publication and its supplements was too sparse, the item was rated unclear. The tools also provide the option “not applicable” for items that are not appropriate for the study. Based on the item ratings, the reviewers assigned an overall ROB rating to each study. The JBI tools do not provide an algorithm for determining the study’s overall ROB rating (low, high, or unclear) based on a scoring system or similar. The overall ROB rating is based on a joint critical appraisal of all relevant aspects of the study, with low methodological study quality in key aspects not being compensated for by other components of the study in which a high methodological quality was present. The evaluation process is inevitably prone to subjective elements. To ensure a consistent rating standard, the reviewers thoroughly reviewed and discussed each item of the JBI tool beforehand and piloted the ROB assessment on two studies to establish specific decision rules for determining the overall ROB.

In case of discrepant ROB ratings, consensus meetings involving a fourth reviewer were held to discuss the disagreements and reach a consensus decision.

### 2.6. Statistical Analysis

The main characteristics of the included studies, their findings, and ROB ratings are presented in summary tables.

The intake of furocoumarins or furocoumarin-containing foods was captured very heterogeneously in the different studies. The studies not only examined a broad spectrum of different furocoumarin-containing foods but also used diverse categorizations of exposure levels and various analytical methods. The transformation of exposure categories to a common scale would most likely have introduced substantial bias because information about the consumption levels of furocoumarin-containing foods in the studies was limited. Therefore, we did not perform a meta-analytical summary of risk estimates. Instead, we performed a qualitative synthesis and summarized the number of studies yielding a positive, negative, or no association between dietary furocoumarin intake and CM. We classified a study as showing a positive association when higher furocoumarin consumption was associated with higher CM risk, a negative association when higher consumption levels were associated with lower CM risk, and no association when the results were inconclusive. In addition to stating how many studies showed positive and negative associations, we also provide information on how many of these were statistically significant based on the statistical evaluation in the study publications.

Observational epidemiological studies assessing the effects of potential CM risk factors typically collect data on UVR exposure to account for its potentially confounding role. Furocoumarin intake, as our exposure of interest, requires, however, special handling of UVR exposure data in the analysis. Rather than just using the data to adjust for confounding, potential effect modification by UVR exposure must be examined because the hypothesized carcinogenic effect of furocoumarins on CM development requires the presence of UVR. Consequently, individuals with higher levels of UVR exposure should exhibit a steeper CM risk gradient for ordered consumption categories of furocoumarin intake compared to those with lower levels of UVR exposure. We investigated whether and how UVR exposure data were used in the analysis. In those publications where potential effect modification has been evaluated, we classified whether the reported results provide no, weak, moderate, or strong evidence for an effect-modifying role of UVR exposure.

## 3. Results

### 3.1. Literature Search

Figure 1 displays the literature search process, which initially yielded 4110 publications. After eliminating 1336 duplicates, the remaining 2774 publications were included in the title and abstract screening, of which 74 publications survived. Subsequent full-text screening excluded a further 56 studies, resulting in 17 studies reported in 18 publications eligible for inclusion in the systematic review. While forward snowballing did not identify additional studies, two more studies were found through the references of the already included studies [20,21]. This finally yielded 20 publications based on 19 distinct studies (Refs. [22,23] reported distinct results from the same study) that entered this systematic review. A list of all publications evaluated in the full-text screening that were excluded, together with the individual reason for exclusion, can be found in Appendix A.

### 3.2. Study Characteristics

Table 2 provides an overview of the study characteristics. The 19 included studies consisted of nine case-control studies, eight cohort studies, one analytical cross-sectional study, and one ecological study (see Table 3).

The studies—published over a wide time interval, between 1986 and 2021—collected information on, altogether, 1,190,209 individuals and contained 13,872 CM cases (corrected for overlap between study populations in [27,32,35], as well as the identity of study populations in [22,23]) from different regions (see Table 3).

### 3.3. Furocoumarin Sources in Studies

The heterogeneity of how and what information on furocoumarin exposure the eligible studies collected and how they reported their data makes comparisons difficult. While the majority of studies (*n* = 15) used food frequency questionnaires for gathering information on the consumption of specific foods, two used 24-h diet recall interviews, and three used either a questionnaire of their own design or the FAOSTAT database (Food and Agriculture Organization Statistics, https://www.fao.org/faostat (accessed on 14 February 2025)). Further, the furocoumarin-rich foods covered were not classified into standardized categories, and the reporting of the consumption frequencies of those categories ranged from specific quantiles of the study data (i.e., tertiles, quartiles, quintiles) to units of food consumed in a certain time period, where “unit” was individually defined in the publications (e.g., pieces/portions of food, glasses with glass sizes measured in either ounces or milliliters and of differing volume), and the time periods vary “per day” and “per week”. A detailed table of the specifics of furocoumarin exposure assessment in each study is presented in the Appendix A.

With the exception of two studies [35,36], all other investigations examined only parts of the furocoumarin-containing food and beverage spectrum and were thus not able to quantify the total level of dietary furocoumarin intake. We classified the information given in these studies into the following furocoumarin categories: “citrus fruits and juices”, “citrus fruits”, “citrus juices”, “grapefruit and grapefruit juice”, “orange and orange juice”, “other citrus fruits”, and “others”. Table 4 presents an overview of the number of publications per category. As one study often covered more than one category of furocoumarin-rich foods, the numbers in Table 4 add up to 45. Detailed quantitative results of the individual studies, including information about the adjustment factors used in the quantitative analyses, can be found in Appendix A.

Citrus fruits and juices as a combined category were investigated in six studies [29,32,34,37,38,39]. Three of them found a positive association [32,38,39], with two of them being significant [32,38]. Meanwhile, [29] found a nonsignificant negative association and [34], as well as [37], none at all. While four of these studies had a low ROB, one study had an unclear ROB [38], and the ROB of one study [29], which showed a negative association, was high.

Another six studies reported mixed results for citrus fruits. Two studies [31,34] found a positive association, with the effect in [34] being significant, one study [30] a significant negative association, and three studies [23,37,39] found no association. The ROB was rated as low in all of these studies.

Citrus juices were covered in three studies. Two studies [34,39] found no association, and one study [37] found a nonsignificant positive association. All studies were rated to have low ROB.

Four studies reported risk estimates for the consumption of grapefruit on the incidence of CM [21,32,38,39] and two for the consumption of grapefruit juice [32,38].

For grapefruit alone, three studies [32,38,39] reported a positive association with the risk of CM, but only one reported a significant effect [32], and one reported no effect [21]. Two of the four studies [32,39] had a low ROB, while one [21] had a high ROB and the remaining one [38] had an unclear ROB.

The two studies covering grapefruit juice reported no association [32] or a nonsignificant positive association [38], though with an unclear ROB in [38].

The consumption of oranges was investigated in four studies [21,26,32,38] and that of orange juice in five studies [21,27,31,32,38]. The only study to establish a positive association between the consumption of oranges and the risk for CM was [38], but the ROB for this study was unclear. The other three, all low ROB studies, found no association. Three out of five studies investigating orange juice reported a significant, positive association with CM incidence [27,32,38], with one [38] having an unclear ROB, while the other two studies had a low ROB. The remaining two low ROB studies did not identify any association [21,31].

The category “other citrus fruits” comprises the results for the following combinations of citrus fruits: “orange, tangerine, tangelo” [39], “mandarin” [38], “orange and grapefruit” [22], “orange and mandarin” [33], “orange juice and grapefruit juice” [22], and “tangerine” [22]. None of these consumption combinations was associated with higher risks of developing CM. The ROB status was high in [33], unclear in [38], and low in the other two studies [22,39].

The last category, “others”, comprises furocoumarin-containing food items not fitting into the above categorization, namely parsley and carrots. No association was found for the consumption of parsley in one study [30], which was assessed as a study with a low ROB. The consumption of carrots was examined in six studies, with two of them reporting a positive association with the risk for CM [28,30] and four reporting no association [20,21,24,25]. Among them, the latter [25] had a high ROB; all others had a low ROB.

The impact of total furocoumarin consumption on the incidence of CM was investigated in two studies [35,36]. While both studies associated higher furocoumarin intake with higher risk for CM, the effects were not significant. Furthermore, one of the studies [25], employing a cross-sectional design and relying on an unvalidated self-reported CM, had a high ROB.

### 3.4. Role of UVR Exposure in the Analysis of the Furocoumarin–Melanoma Relationship

Sixteen of our 20 studies [20,21,22,23,24,27,28,29,30,31,32,34,35,37,38,39] collected some UVR data and used it in their analyses; only four [25,26,33,36] did not. Of the seven studies [32,34,35,36,37,38,39] specifically evaluating the furocoumarin–melanoma association, six [32,34,35,37,38,39] reported the results of statistical analyses evaluating whether UVR exposure acts as an effect modifier of the association under study. Five of these studies [32,34,35,37,39] used stratification by UVR exposure and analyzed the Cox models in two subgroups defined by UVR exposure to evaluate the effect-modifying role of UVR exposure (two studies [37,39] reported results for the two UVR subgroups without stating whether the apparent differences were significant; three studies [32,34,35] reported results for the UVR subgroups and stated that the differences between subgroups were not significant). One study [38] incorporated a product term between total citrus consumption and average outdoor time in summer as a proxy for UVR exposure in a logistic regression model, tested statistically for interaction using the likelihood ratio test, and stated that there was no significant interaction without giving further results. None of the other 13 studies [20,21,22,23,24,25,27,28,29,30,31,33], which did not focus on the relationship between furocoumarin intake and CM and only reported data on some furocoumarin-containing foods and/or beverages and their association with CM risk as a byproduct, considered the role of UVR exposure as a potential effect modifier. Notably, none of the six studies evaluating UVR exposure as a potential effect modifier did so in their primary analyses. In all of these cases, the primary analyses adjusted for a confounding effect of UVR exposure.

The assessment of UVR exposure and reporting of the results in the six studies investigating the role of UVR exposure in the furocoumarin–melanoma association were heterogeneous. The reported results in four studies [32,35,37,39] showed moderate-to-strong indications of an effect-modifying role of UVR exposure. However, the statistical significance of a test for interaction was not claimed in these cases. In all of these studies, the subgroups with high UVR exposure showed a steeper gradient of CM risk depending on the level of furocoumarin intake (total furocoumarin intake in [35], citrus fruits and citrus juice in [37], grapefruits in [32], and citrus fruits and oranges/tangerines/tangelos in [39]) compared to the low UVR exposure subgroups. In all four studies, statistical trend tests addressing a dose–response effect of furocoumarin intake on CM were statistically significant only in the high UVR exposure subgroup, not in the low UVR exposure group. One study [34] showed only weak evidence for an effect-modifying role of UVR exposure, only for citrus fruits, but not for citrus juice and total citrus intake. A slightly steeper CM risk gradient for consumption quartiles was observed in the subgroup with more recreational physical activity in summer (as an imprecise proxy measure for UVR exposure used in this study). Interestingly, only three studies [32,34,37] explicitly pointed to a potential synergistic interaction between furocoumarins and UVR in their Discussion section, while the others neglected this aspect.

## 4. Discussion

The potential impact of dietary furocoumarin intake on CM development has attracted considerable scientific interest in recent years. Furocoumarins have well-known carcinogenic properties, as established in experimental studies in mice [40] and epidemiological studies in humans [15]. Topically applied furocoumarins have a photosensitizing effect on the skin and, in interaction with UVR, stimulate the proliferation of melanoma cells [41]. Whether the dietary intake of furocoumarins has a similar effect is currently a matter of debate. In our systematic review, we compiled the complete epidemiological literature on that topic. After an in-depth literature search, we found 20 publications based on 19 different studies that provided information on the furocoumarin–melanoma association. We refrained from a meta-analytical synthesis of the results of these studies because of the large heterogeneity in exposure assessment, operationalization of furocoumarin intake in the analyses, and analytical methods of the studies. Instead, we provided a comprehensive overview of the studies and a qualitative summary of their findings. Overall, we found moderate evidence supporting the notion that dietary furocoumarin intake at higher levels acts as a risk factor for CM.

Our systematic review constitutes the first qualitative summary of the full spectrum of epidemiological studies on this topic. A recent dose–response meta-analysis limited to five cohort studies and pragmatically transforming exposure categories used in the individual studies into a common scale established a significant linear relationship between CM risk and the consumption of citrus fruits and juices [42]. This finding is in line with the qualitative results of our systematic review. However, our approach considered all epidemiological study types on the topic and discussed the large heterogeneity of the individual methods and results, which the meta-analysis had to pragmatically simplify. In addition, our systematic review also analyzed in detail the role of UVR exposure as an effect modifier of the furocoumarin–melanoma relationship, which the meta-analyses did not address.

The interpretation of the results from the epidemiological studies on dietary furocoumarin intake on CM risk requires caution. In order to make a methodologically sound and quantitative statement about the effect of furocoumarins on CM risk, we would need to collect more detailed data on the amount of furocoumarins ingested from a given food in a given time window and the level of UVR exposure in the same time window from the study participants. Such data are difficult to obtain in a practical study and were not available in all of the studies covered by this systematic review. Dietary habits were self-reported and mostly assessed using food frequency questionnaires, without any detailed information on variations in food consumption over time. The latter is relevant when considering the furocoumarin–UVR interaction. In the northern hemisphere, e.g., the winter months, during which UVR is at a low level, are the high season for the intake of citrus fruits. Citrus fruit consumption during low UVR periods does not pose a risk for CM development.

Another issue that made the synthesis of study results difficult was the assessment, as well as the reporting, of intake quantities and frequencies of specific foods. While some studies specified portion size references precisely, e.g., one or half a fruit and one glass of 177.5 mL/250 mL, others defined portions sizes more vaguely as, e.g., a “medium-sized portion”, and again, others had participants estimate their intake in grams, used pictures to decide portion sizes, or did not specify them at all. In terms of consumption frequency, the captured range comprised “number of portions per month”, “per week”, and “per day”. Often, the studies reported results in the categories the exposure was captured, while others estimated the weight of portion sizes, categorized the original portion sizes into “low/medium/high consumption”, or subdivided the consumption levels into quantiles (tertiles, quartiles, and quintiles).

A further complicating issue relates to the food categories used by the included studies when reporting their results. Apart from a few individually stated foods (grapefruit, grapefruit juice, orange, orange juice, parsley, and carrot) many studies reported results on combinations of consumed goods (citrus fruits and juices, citrus fruits, citrus juices, and other citrus fruits), with only some stating exactly what they summarized in these combinations and others not at all. These broad categories do not allow the identification of individual food item risks. Another aspect that complicates the comparison is a circumstance that affects all exposure categories, namely that the furocoumarin content of the same food item is subject to high variability due to a variety of causes. These causes include the cultivar of a particular food, the growing conditions and geographical location, the degree of ripeness at harvest and possible post-ripening, the storage conditions, and the processing, especially heat treatment [12].

Focusing on specific food or beverage items, as has been done in most studies, bears the risk of confounding because there may be a positive or negative association in the consumption levels of furocoumarin-containing food or beverage items that is not adjusted for in analyses investigating only a specific food or beverage item. Another approach that is better suited to quantify the magnitude of the furocoumarin–melanoma association is the use of the accumulated total furocoumarin intake, covering the whole spectrum of furocoumarin-containing food. This approach was adopted by two studies [35,36] determining the cumulative dietary intake of furocoumarins based on a database of furocoumarin content in foods developed by Melough et al. [43]. Despite being subject to the same limitation resulting from the variability of furocoumarin levels as specific food items, this procedure allows for a determination of the risk for CM related to the photosensitizing substance itself.

We observed evidence in our synthesis that the consumption of grapefruits stands out in the sense that nearly all studies reporting effects for grapefruits reported positive associations with the risk of CM, whereas this is not true for the consumption of grapefruit juice. This apparent contradiction can be explained by the different furocoumarin content in both. Fresh grapefruit contains more than twice as much furocoumarin as grapefruit juice [43]. Wu et al. [32] reported that the effect of temperature during juice processing reduces the furocoumarin content. In addition, fresh grapefruits contain substantially higher levels of furocoumarin when compared to other citrus fruits [43]. This finding offers a probable explanation for the observed association between grapefruit consumption and melanoma risk, which is notably more prominent than that associated with other furocoumarin-containing foods.

It should also be noted that the methodological approach to handling variables related to UVR exposure in the analysis requires reconsideration. Furocoumarins are photosensitizing substances, and their carcinogenic effects are triggered only by UVR. Activated by the absorption of UVR energy, furocoumarins can interact with the DNA in skin cells, which fosters the development of DNA damage and mutations. However, without UVR exposure, this reactive property remains inactive [12]. With this knowledge, it is evident that UVR exposure must be considered as an effect modifier when modeling the relationship between the consumption of furocoumarins and CM risk and not as a confounder. However, most studies included UVR exposure only as a confounder in their analyses.

Although not all six studies that considered UVR exposure as a potential effect modifier in their analyses showed a stronger association between dietary furocoumarin consumption and CM risk in subgroups with higher UVR exposure, four studies provided evidence for the effect-modifying role of UVR exposure. The heterogeneous results may be due to a suboptimal quantification of UVR exposure. For example, the publication by Mahamat-Saleh et al. [34], which reported only weak evidence for an effect-modifying role of UVR exposure, used the total hours of recreational physical activity in summer as a proxy for recreational sun exposure. It is questionable whether this adequately reflects actual sun exposure, as even the authors themselves concede in their discussion section.

Furthermore, a direct temporal link between the consumption of furocoumarin-containing foods and UVR exposure is required for a potential effect, as the photosensitizing effect of furocoumarins is only present for a limited period of time. Several studies have shown that skin furocoumarin concentrations reach their peak a few hours after oral consumption and then decline rapidly [44,45]. How long the furocoumarins remain in the skin tissue depends on the type of furocoumarins, the administered dose, and the individual’s metabolism. Consequently, only UVR exposure directly after the consumption of a furocoumarin-containing food is relevant when considering UVR exposure as an effect modifier. Nevertheless, such a detailed assessment of UVR exposure is challenging.

A further potential explanation for the ambiguous results regarding the effect-modifying role of UVR exposure in previous studies relates to the lack of research conducted in regions characterized by elevated UVR levels. The existing literature on this subject is predominantly derived from studies conducted in the USA and in European countries like the UK. Thus, the incorporation of data from a region with a year-round high UVR level, such as Australia, which is renowned for its extensive research on melanoma, could offer novel insights into the subject.

We acknowledge that this systematic review has some limitations. It is possible that our search string did not identify all relevant research on this topic. Nevertheless, we implemented a comprehensive search strategy that included database searching, forward and backward citation tracking, and also consultation with dermato-oncological experts, in order to minimize the likelihood of overlooking relevant research. Due to the language restriction to English and German, it cannot be ruled out that publications in other languages were missed. However, this is considered to be very unlikely, as only a single publication was excluded during the full-text screening due to a foreign language. As described in the methods, we used JBI’s checklists to assess the ROB of the included studies in a structured and transparent manner. However, we deviated from this approach in the study of Grasgruber et al. [33], as there is no checklist for ecological studies. Nevertheless, we thoroughly discussed this study within the group and unanimously rated it as a high ROB study. Furthermore, we cannot rule out the theoretical possibility of residual confounding explaining the observed furocoumarin-melanoma association. As the included studies made appropriate efforts to control confounding and reported adjusted risk estimates, we do not think that this is a likely explanation.

## 5. Conclusions

In conclusion, our systematic review, comprising results from 20 publications, could not provide a definite answer to the research question of whether and to what extent dietary furocoumarin intake increases melanoma risk. We found moderate evidence that higher dietary furocoumarin intake acts as a risk factor for CM, but the heterogeneity of approaches and shortcomings in adequately incorporating UVR exposure data in the analysis of the studies precluded a more far-reaching statement. However, according to the precautionary principle, one could be cautious about consuming high amounts of furocoumarin-rich foods (such as grapefruit) during periods with high UVR and prolonged outdoor stays.

Future epidemiological analyses of the furocoumarin–melanoma association require more comprehensive dietary and UVR exposure data to better characterize the individual total furocoumarin intake and its interplay with individual UVR exposure patterns.

## Figures and Tables

**Figure 1 nutrients-17-01296-f001:**
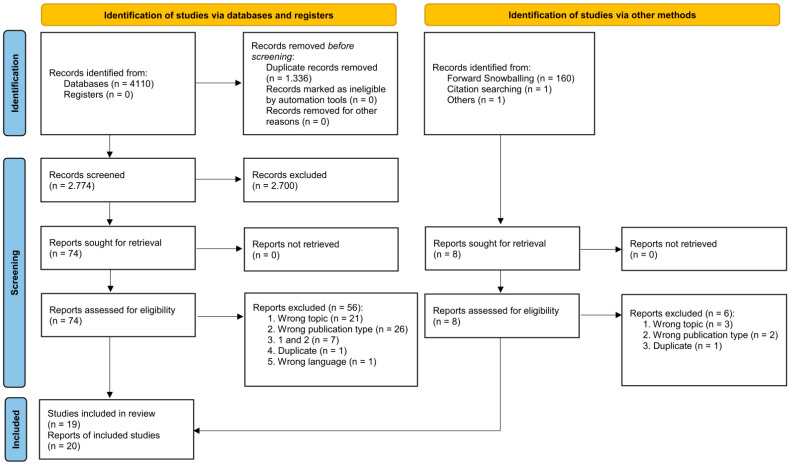
PRISMA flowchart with details of the search process.

**Table 1 nutrients-17-01296-t001:** Eligibility criteria according to the PECOS scheme.

Aspect	Eligibility Criteria
Population	Animal studies were excluded. No further restrictions regarding human study populations.
Exposure	Consumption of furocoumarin-containing foods, such as fig, carrot, parsley, turnip, celery, dill, coriander, cumin, citrus fruits (lemon, lime, grapefruit, orange or tangerines (mandarin and clementine), and beverages containing furocoumarins such as carrot juice, orange juice, lemon juice, lime juice, and grapefruit juice
Comparison	Other human populations with different exposure levels
Outcome	Development of cutaneous melanoma
Study design	Observational studies, i.e., cohort studies, case-cohort studies, (nested) case-control studies, analytical cross-sectional studies, and ecological studies

**Table 2 nutrients-17-01296-t002:** Summary of included studies (chronologically ordered according to publication year).

First Author of Publication and Reference	Recruiting Period	Country	Study Type	Sample Size (Cases)	Foods and Food Combinations Investigated	ROB
Holman [20]	January 1980–November 1981	Australia	case-control	1022 (511)	carrot	low
Østerlind [24]	1982–1985	Denmark	case-control	1400 (474)	carrot	low
Stryker [25]	July 1982–August 1985	USA	case-control	452 (204)	carrot	low
Soliman [21]	February 1987–January 1992	USA	case-control	873 (261)	Grapefruit ^†^, carrot, orange juice, orange	low/high ^†^
Veierød [26]	1977–1983	Norway	cohort	50,757 (108)	orange	low
Feskanich [27]	1980–1998	USA	cohort	162,078 (414)	orange juice	low
Naldi [28]	1992–1994	Italy	case-control	1080 (542)	carrot	low
Millen [29]	1991–1992	USA	case-control	1058 (497)	citrus fruits and juices	high
Fortes [30]	May 2001–May 2003	Italy	case-control	609 (304)	citrus fruits (orange, mandarin), parsley, carrot	low
Vinceti [31]	not reported	Italy	case-control	118 (59)	citrus fruits	low
Malavolti ^‡^ [22]	2005–2006	Italy	case-control	1099 (380)	tangerine, orange and grapefruit, orange juice and grapefruit juice	low
Wu * [32]	NHS 1984–1998, HPFS 1986–1998	USA	cohort	105,432 (1840)	citrus fruits and juices (grapefruit, grapefruit juice, orange, orange juice), grapefruit, grapefruit juice, orange, orange juice	low
Grasgruber [33]	1993–2011	Europe	Ecological	- ^§^	orange and mandarin	high
Malagoli ^‡^ [23]	2005–2006	Italy	case-control	1099 (380)	citrus fruits	low
Mahamat-Saleh * [34]	1992–2000	Europe	cohort	270,112 (1371)	citrus fruits and juices, citrus fruits, citrus juices	low
Sun * [35]	NHS 1984–1998, HPFS 1986–1998	USA	cohort	122,744 (1593)	total furocoumarin consumption	low
Melough * [36]	2003–2012	USA	cross sectional	11,696 (75)	total furocoumarin consumption	high
Melough * [37]	1993–1998	USA	cohort	56,205 (956)	citrus fruits and juices (orange, grapefruit, tangerine, orange juice, grapefruit juice), citrus fruits (orange, grapefruit, tangerine), citrus juices (orange juice, grapefruit juice)	low
Marley * [38]	2006–2010	UK	cohort	198,964 (1592)	citrus fruits and juices (grapefruit, grapefruit juice, mandarin, orange, orange juice), grapefruit, grapefruit juice, mandarin, orange, orange juice	unclear
Melough * [39]	1995–1996	USA	cohort	388,467 (3894)	citrus fruits and juices (grapefruits, orange, tangerine, tangelo, orange and grapefruit juice), citrus fruits (grapefruits, orange, tangerine, tangelo), citrus juices (orange and grapefruit juice), grapefruit, orange/tangerine/tangelo	low

NHS = Nurses’ Health Study, HPFS = Health Professionals Follow-up Study; * Studies explicitly investigating furocoumarins. ^†^ High ROB only for the grapefruit–melanoma association because no adjusted risk estimates are reported for grapefruits. ^‡^ Both publications refer to the same case-control study. ^§^ No sample size information because no individual data are gathered in an ecological study.

**Table 3 nutrients-17-01296-t003:** Study design, location, and publication time of the 19 distinct studies included in the systematic review.

	N (n) *	%
Type of study		
case-control studies	9 (0)	47.4
cohort studies	8 (6)	42.1
cross-sectional studies	1 (1)	5.3
ecological studies	1 (0)	5.3
Geographic region		
USA	9 (5)	47.4
Italy	4 (0)	21.1
Europe	2 (1)	10.5
Australia	1 (0)	5.3
Denmark	1 (0)	5.3
Norway	1 (0)	5.3
UK	1 (1)	5.3
Publication period ^†^		
before 1990	3 (0)	15.0
1990–1999	2 (0)	10.0
2000–2009	5 (0)	25.0
2010–2019	4 (1)	20.0
2020 and later	6 (6)	30.0

* Absolute number of studies (numbers in parentheses refer to the absolute number of studies specifically addressing the effect of dietary furocoumarins); ^†^ Numbers add up to 20 since one study was published twice in two different time categories.

**Table 4 nutrients-17-01296-t004:** Summary of publications per category and their type of associations.

	Publications	Association		
	n *	n (n_sig_)		
Furocoumarin Food/Beverage Category		+	o	−
Citrus fruits and juices	6	3 (2)	2	1 (0)
Citrus fruits	6	2 (1)	3	1 (1)
Citrus juices	3	1 (0)	2	
Grapefruit and grapefruit juice				
Grapefruit	4	3 (1)	1	
Grapefruit juice	2	1 (0)	1	
Oranges and orange juice				
Orange	4	1 (1)	3	
Orange juice	5	3 (3)	2	
Other citrus fruits				
Orange, tangerine, tangelo	1		1	
Mandarin	1		1	
Orange and grapefruit	1		1	
Orange and mandarin	1		1	
Orange juice and grapefruit juice	1		1	
Tangerine	1		1	
Others				
Parsley	1		1	
Carrot	6		4	2 (2)
Total furocoumarin consumption	2	2 (0)		

* Due to some studies investigating several furocoumarin food/beverage categories simultaneously, the number of studies adds up to 45. n: number of studies, n_sig_: number of studies reporting associations with *p*-trend/*p* < 0.05, +: positive association, o: no association, −: negative association.

## Data Availability

The data are included in the article.

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
