# Peer review of "The Impact of Dietary Intake of Furocoumarins and Furocoumarin-Rich Foods on the Risk of Cutaneous Melanoma: A Systematic Review"

_nutrients, 2025, doi:10.3390/nu17081296_

Round 1

Reviewer 1 Report

Comments and Suggestions for Authors

Please, see the attached document. 

This systematic review by Kaiser et al. synthesizes epidemiological studies on the association between diet furocoumarin exposure and cutaneous melanoma risk, combining 19 studies with design and measurement heterogeneity. While there is some evidence of risk increase with increased furocoumarin exposure, particularly from grapefruit intake, overall findings are inconclusive due to pervasive heterogeneity among studies and thin confounding adjustment. Future research must integrate complete dietary and UV exposure assessments to establish the contribution of dietary furocoumarins to melanoma risk. There are a number of methodological, linguistic, formal, and scientific concerns that must be resolved prior to acceptance for publication. The discussion below deals with these concerns:

  1. Why was a quantitative synthesis not even tried, e.g., stratification by subgroup or sensitivity analysis? Please, clarify the lack of meta-analysis more and attempt a quantitative synthesis.
  2. How were discrepancies between reviewers in data extraction and study selection resolved?
  3. Were confounders other than UV exposure accounted for, e.g., skin type or genetic susceptibility factors? Please, state whether interaction tests were performed and how UV exposure data were incorporated.
  4. Would you describe how UVR exposure was investigated as an effect modifier? Were interaction terms defined in any included studies? Could you please note whether interaction tests were conducted and how UV exposure data were incorporated?
  5. Would you describe why grapefruit consumption always showed stronger associations compared to other citrus fruits?
  6. With the vast heterogeneity between studies, how certain are you that the results are due to furocoumarin intake rather than residual confounding?
  7. Can more prospective cohort studies with higher exposure measurement accuracy fill the identified gaps?
  8. Clinicians and the general public--how are they supposed to be able to utilize these findings in their dietary advice?
  9. Kindly, revise the text to make it concise, edit grammatical mistakes, and enhance structure, edit formatting errors and provide missing citations.

Comments on the Quality of English Language

Please, see the attached document. 

Reviewer 2 Report

Comments and Suggestions for Authors

The major revision Comments in this work concentrate on the subsequent essential aspects need to modify:

Significant diversity exists in the assessment of furocoumarin consumption among research, complicating quantitative synthesis of findings.

A meta-analysis was not performed due to variations in study designs and measurement techniques. A qualitative synthesis was presented alternatively. The author need to explain it.

The analysis insufficiently included UV radiation exposure as a modifying component, essential for comprehending furocoumarin's impact on melanoma risk.

Several featured research provided limited data on dietary furocoumarin consumption, limiting the capacity to reach definitive findings.

The classification of furocoumarin-containing foods differed across research, resulting in difficulties in comparing outcomes.

Most studies failed to report cumulative dietary furocoumarin intake, concentrating instead on particular items, potentially introducing confounding variables. Need to clarify it.

Subsequent research should focus on gathering more accurate data regarding furocoumarin intake and ultraviolet exposure to enhance risk assessment. Please add more recent information regarding it.

Round 2

Reviewer 1 Report

Comments and Suggestions for Authors

After revisions I recommend for publication. 

Author Response

A reply is not required.

Reviewer 2 Report

Comments and Suggestions for Authors

no further comments

Author Response

A reply is not required.